# The Prevalence of Human Papillomavirus (HPV) Genotypes in the Oral Mucosae of HIV-Positive Individuals: A Systematic Review and Meta-Analysis

**DOI:** 10.3390/microorganisms13030646

**Published:** 2025-03-12

**Authors:** Gul Bayram, Tugce Simsek Yildirim, Elif Ertas, Arzu Kanik

**Affiliations:** 1Department of Medical Services and Techniques, Vocational School of Health Services, Mersin University, Mersin 33160, Türkiye; 2Tugce Simsek Private Clinic, Sisli, İstanbul 34370, Türkiye; info@docdrtugcesimsek.com; 3Department of Biostatistics, Selcuk University, Konya 42130, Türkiye; eelifertass@gmail.com; 4Department of Biostatistics and Medical Informatics, Faculty of Medicine, Mersin University, Mersin 33343, Türkiye; arzukanik@mersin.edu.tr

**Keywords:** human papillomavirus, oropharyngeal, genotypes, HIV-positive

## Abstract

Papillomaviruses are double-stranded DNA viruses, and it is essential to clarify their genotypic distribution for their effective prevention and clinical management. In this study, we aimed to evaluate the prevalence of HPV genotypes in the normal oral mucosae of HIV-positive individuals. A systematic literature search was conducted across PubMed, Web of Science, Scopus, and Google Scholar to identify peer-reviewed studies published up to 13 February 2025. The inclusion criteria referred to original research studies reporting on the prevalence and genotype-specific distribution of HPV in the oral mucosae of HIV-positive individuals. Statistical analyses were conducted using the MedicReS E-PICOS AI smart biostatistics software (version 21.3, New York, NY, USA) and the MedCalc statistical software package (MedCalc Software Ltd., Ostend, Belgium). The pooled prevalence estimates were calculated using a random-effects meta-analysis model, and heterogeneity was quantified using the Cochrane Q and I^2^ statistics. The presence of publication bias was assessed via the Begg and Mazumdar rank correlation test. High prevalence and heterogeneity of HPV-58 (6.23%), HPV-16 (4.326%), and HPV-66 (3.733%) were observed, indicating significant variability across populations and methodologies. This supports their association with HPV-related oropharyngeal malignancies and the need for the continuous surveillance of HIV-positive individuals. We also observed the elevated detection of LR-HPV genotypes, particularly HPV-13 (7.16%), HPV-5 (5.64%), and HPV-62 (4.24%). These findings indicate that there is substantial heterogeneity in the prevalence of both HR-HPV and LR-HPV genotypes among HIV-positive individuals, with certain genotypes exhibiting higher detection rates across studies, emphasizing the need for targeted surveillance and preventive strategies in this vulnerable population. The application of advanced data analysis methods is essential in enhancing HPV surveillance and implementing effective control measures in this vulnerable population.

## 1. Introduction

Papillomaviruses are double-stranded, icosahedral DNA viruses characterized by significant genetic heterogeneity [1]. These viruses are recognized for their diverse clinical and pathological implications across various hosts and tissue types. Papillomaviruses primarily infect the epithelial tissue of vertebrates, either causing neoplasia or persisting asymptomatically. Papillomaviruses are classified into four major genera: *Alphapapillomavirus*, *Betapapillomavirus*, *Gammapapillomavirus*, and *Deltapapillomavirus* [2]. More than 200 different genotypes have been identified. The oncogenic potential of papillomaviruses varies significantly across these genera. HPV is divided into two main groups—low-risk and high-risk types—based on their association with cancer [3]. *Alphapapillomavirus* includes high-risk types such as HPV-16 and -18, which are reported to be the primary causes of cervical cancer, whereas low-risk types (e.g., HPV-6 and -11) cause benign warts [4]. *Betapapillomavirus* is associated with non-melanoma skin cancers, particularly in immunosuppressed individuals. *Gammapapillomavirus* primarily causes benign cutaneous lesions, while *Deltapapillomavirus* affects animals and is not known to cause cancer in humans. These distinctions highlight the clinical significance of papillomaviruses in human health, particularly regarding their role in disease pathogenesis and the epidemiological dynamics of viral transmission [5]. Moreover, HPV-related infections are noted to have significant implications, not only in terms of cervical cancer in women, but also in the etiology of oropharyngeal cancers and other mucosal pathologies. According to estimates, HPV infections are responsible for 99% of all cervical cancers, 90% of all anal cancers, 69% of all vulvar cancers, 75% of all vaginal cancers, 40% of all penile cancers, and 70% of all oropharyngeal cancers. These data indicate that HPV is one of the most common viral agents responsible for cancer worldwide [3]. Currently, global health authorities prioritize HPV vaccination programs and population-based screening initiatives for cervical cancer prevention [6,7,8]. Although HPV vaccination programs and population-based screening initiatives have significantly reduced the incidence of cervical cancer, the role of HPV in oropharyngeal squamous cell carcinoma (OPSCC) remains underexplored. Studies on the etiology of oropharyngeal squamous cell carcinoma (OPSCC) remain limited, but the presence of high-risk HPV types in the oral mucosae during the early stages has been widely observed, particularly in vulnerable and immunosuppressed patient groups, such as HIV-positive individuals [9,10,11]. A growing body of evidence suggests that HPV-related oropharyngeal infections are particularly concerning in HIV-positive individuals, who exhibit a higher prevalence of HPV and increased persistence due to their compromised immune responses. The persistence of HPV in immunocompromised individuals can increase the likelihood of progression to malignancy. Although previous studies have described the prevalence of HPV in cervical and anogenital sites, research focusing on oral HPV infections in HIV-positive individuals remains insufficient. Given the higher viral persistence, increased genotypic diversity, and potential for malignant transformation in this population, it is crucial to better characterize HPV’s epidemiology, transmission dynamics, and clinical outcomes in terms of the oral mucosae of HIV-positive individuals. Due to the weakening of the immune system, HPV infections are not only more prevalent in this group, but also tend to persist for longer periods, leading to broader genotypic diversity compared to the general population [12,13,14,15]. Immune-related adverse events (irAEs) have emerged as a significant concern in immunocompromised individuals undergoing various therapeutic interventions. A recent study investigating the incidence of noninfectious uveitis following immune checkpoint inhibitor (ICI) therapy has underscored the profound impact of immune modulation on disease susceptibility among vulnerable populations. Likewise, HPV-related infections in HIV-positive individuals represent a major clinical challenge due to immune system dysregulation, which leads to increased viral persistence and higher prevalence rates. Elucidating the complex interplay between immune dysfunction and disease progression in these contexts is essential in advancing our understanding of viral oncogenesis and immune surveillance mechanisms. This highlights the need for a more comprehensive understanding of HPV’s epidemiology and transmission dynamics and the clinical outcomes associated with different HPV types, particularly in vulnerable populations such as HIV-positive individuals [16].

Based on the above, in this study, we aimed to evaluate the prevalence and genetic diversity of HPV genotypes in the normal oral mucosae of HIV-positive individuals, thereby addressing a significant gap in the existing literature and contributing novel insights into the epidemiological patterns and clinical implications of HPV in immunocompromised populations.

## 2. Materials and Methods

This review was conducted using the Preferred Reporting Items for Systematic Reviews and Meta-Analysis (PRISMA 2020, [17]) Guidelines (Figure 1). The systematic review protocol was registered in PROSPERO (Registration No.: CRD42025638316). The review process was carried out between 1 January 2025 and 13 February 2025. PubMed, Web of Science, Scopus, and Google Scholar were used as search engines. The primary keywords used for the literature search were “oropharyngeal HPV HIV”, yielding 19,400 articles in Google Scholar, 191 in PubMed, 158 in Scopus, and 209 in Web of Science (WoS). To limit the search results to HIV-positive individuals, the phrase “oropharyngeal HPV in HIV positive” was applied, resulting in 16,700 articles in Google Scholar, 77 in PubMed, 57 in Scopus, and 76 in WoS. Each included study was manually reviewed to ensure its relevance, with only those that described genotyping within high- and low-risk HPV cases being included. After removing duplicates and manually screening the titles and abstracts, 20 studies remained for full-text review. These were further screened based on specific inclusion and exclusion criteria. In particular, studies focusing on the prevalence of HPV genotypes in the oral mucosae of HIV-positive patients were included. Studies were excluded for the following reasons: (1) a lack of specific HPV genotype data, reporting only high-risk (HR) or low-risk (LR) classifications; (2) the inclusion of patients with head and neck cancer or oral lesions, which could have confounded the results as they do not reflect the normal oral mucosa; (3) the use of case reports or reviews or non-English publications; and (4) a lack of full-text access. A total of 20 studies investigating the prevalence of HPV genotypes in the normal oral mucosae of HIV-positive patients were included in the analysis (Figure 1).

### Statistical Methods

The effect size for HPV was assessed using proportions. The meta-analysis of these proportions was conducted using the random-effects model, where weighted coefficients and proportions with 95% confidence intervals (CIs) were used. To determine the presence of publication bias among the included studies, the Begg and Mazumdar rank correlation test was performed. A forest plot was presented with a 95% CI, where the marker size was adjusted according to the study’s weight.

Heterogeneity was assessed using the Cochrane Q statistic, also known as the chi-squared heterogeneity test, with (k−1) degrees of freedom. The degree of heterogeneity was quantified using the I^2^ statistic. Based on the heterogeneity results, statistical significance was evaluated using both the fixed-effects model and the random-effects model. For data analysis, the MedicReS E-PICOS AI smart biostatistics software (version 21.3, New York, NY, USA) and the MedCalc v23.0.2 statistical software package (MedCalc Software Ltd., Ostend, Belgium) were used.

## 3. Results

### 3.1. Risk of Bias Assessment

For both the HR-HPV and LR-HPV groups, publication bias was assessed using Begg’s test. For the HR-HPV group, the calculated *p*-value was 0.79 (*p* > 0.05), indicating that there was no significant publication bias in the analysis. Additionally, the Kendall’s Tau value was 0.042, supporting the absence of publication bias. Similarly, for the LR-HPV group, the *p*-value was 0.65 (*p* > 0.05), suggesting no significant publication bias. The Kendall’s Tau value for the LR-HPV group was 0.074, again confirming the lack of bias in this analysis.

### 3.2. Heterogeneity Analysis

Heterogeneity was assessed using the Q statistic, *p*-value, and I^2^ statistic. For the HR-HPV group, the Q statistic was 478.644 and the *p*-value was <0.001, indicating that there was significant heterogeneity (*p* < 0.05). Likewise, for the LR-HPV group, the Q statistic was 578.94 and the *p*-value was <0.001, indicating significant heterogeneity (*p* < 0.05). The I^2^ value was 95%, suggesting a high level of heterogeneity (Figure 2). Due to this substantial heterogeneity, a random-effects model was used in the meta-analysis.

### 3.3. HPV Prevalence Results

In this study, by employing a random-effects model, we found the overall HPV prevalence to be 27.97%, with a 95% confidence interval (CI) ranging from 19.34% to 37.51%. This was based on a total sample size of 2430 participants (Table 1). The results demonstrated that there was considerable heterogeneity in the prevalence estimates across the included studies, with the proportions varying from 3.53% (95% CI: 1.71–6.41) in Gaester et al. to 67.84% (95% CI: 61.73–73.53) in Gonçalves et al.) [13,18] (Table 2). This substantial variation underscores the presence of significant methodological and population-level differences among the studies. The relative study weights ranged from 4.59% to 5.23%, with studies with a larger sample size exerting a greater influence on the pooled estimate. In this study, publication bias in the prevalence of HR-HPV genotypes was evaluated using Kendall’s Tau correlation coefficient and Begg’s test. The results indicated that, for all HR-HPV genotypes, the *p*-values exceeded the significance threshold of 0.05 (*p* > 0.05), suggesting the absence of statistically significant publication bias. Both fixed-effects and random-effects models were employed to obtain estimates for the different HR-HPV genotypes (Figure 3). Using the fixed-effects model, it was found that the prevalence rates for HPV-82, HPV-38, HPV-73, HPV-33, HPV-51, HPV-68, HPV-69, HPV-32, HPV-31, and HPV-53 ranged from 1.04% to 2.745%. The relatively narrow confidence intervals (CIs) for these genotypes indicated that there was less heterogeneity among the included studies. Conversely, the random-effects model was used to estimate the prevalence rates for HPV-35, HPV-58, HPV-66, HPV-18, HPV-16, HPV-56, HPV-45, HPV-39, and HPV-59, which ranged from 1.946% to 6.23%. The wider confidence intervals and higher heterogeneity values observed for these genotypes suggest substantial variability across the studies. In particular, HPV-58 (6.23%, 95% CI: 2.773–10.597), HPV-16 (4.326%, 95% CI: 2.391–6.797), and HPV-66 (3.733%, 95% CI: 1.016–8.063) exhibited notably high heterogeneity. These findings highlight the potential variability in the prevalence of HR-HPV genotypes among HIV-positive individuals across different populations and when different methodologies are applied.

In this meta-analysis, the overall prevalence of LR-HPV genotypes in HIV-positive individuals was estimated to be 26.65%, with a 95% confidence interval (CI) ranging from 17.43% to 37.04%, indicating considerable variability across the studies (Table 3). The highest LR-HPV prevalence was reported by Gonçalves et al. (67.84%; 95% CI: 61.73–73.53), followed by Garcia et al. (55.67%; 95% CI: 45.23–65.76) and Chuerduangphui et al. (43.64%; 95% CI: 34.20–53.42) [18,30,31]. In contrast, the lowest prevalence rates were observed in Gaester et al. (3.53%; 95% CI: 1.71–6.40), Santosh et al. (8.33%; 95% CI: 1.75–22.47), and Tsikis et al. (8.62%; 95% CI: 2.86–18.98) [13,22,29]. This substantial variation in the prevalence rates suggested that there was high heterogeneity among the studies, likely due to differences in the HPV detection methods used, regional and demographic variations, and differences in the immunosuppression status among HIV-positive individuals.

The prevalence of low-risk (LR) HPV genotypes in HIV-positive individuals was analyzed using both fixed-effects and random-effects models. The results demonstrated that there was substantial variation in the prevalence rates across different LR-HPV genotypes (Figure 4). Using the random-effects model, it was found that the prevalence of HPV-13 was 7.16% (95% CI: 0.76–19.44), that of HPV-70 was 2.97% (95% CI: 0.80–6.45), that of HPV-61 was 3.02% (95% CI: 0.83–6.51), that of HPV-6 was 3.33% (95% CI: 1.55–5.75), and that of HPV-11 was 2.57% (95% CI: 1.17–4.49). The strong variability in the confidence intervals suggested that there was notable heterogeneity among the studies, likely influenced by differences in the sample populations and detection methods. Using the fixed-effects model, the prevalence of several LR-HPV genotypes was also evaluated. HPV-62 had a prevalence of 4.24% (95% CI: 1.95–7.11), while that of HPV-89 was estimated at 2.07% (95% CI: 0.74–4.30) and that of HPV-72 at 1.87% (95% CI: 0.95–3.28). Moreover, HPV-132 (1.62%; 95% CI: 0.31–4.81), HPV-120 (1.82%; 95% CI: 0.05–9.72), HPV-145 (1.66%; 95% CI: 0.32–4.92), and HPV-5 (5.64%; 95% CI: 2.72–10.18) were also identified. Additional results obtained using the fixed-effects model indicated that HPV-82 had a prevalence of 1.78% (95% CI: 0.48–4.55), that that of HPV-85 was 0.91% (95% CI: 0.32–2.03), that of HPV-83 was 2.57% (95% CI: 1.17–4.49), that of HPV-44 was 0.87% (95% CI: 0.27–2.06), that of HPV-67 was 1.12% (95% CI: 0.40–2.48), and that of HPV-72 was 1.87% (95% CI: 0.95–3.28). Additionally, the prevalence of HPV-84 was found to be 1.76% (95% CI: 0.82–3.04), that of HPV-81 was 2.08% (95% CI: 1.10–3.56), that of HPV-55 was 3.12% (95% CI: 1.84–4.93), that of HPV-42 was 0.70% (95% CI: 0.12–2.24), and that of HPV-54 was 1.66% (95% CI: 0.61–3.60). A high degree of heterogeneity was observed in the prevalence of low-risk (LR) HPV genotypes among HIV-positive individuals, with certain genotypes exhibiting significantly higher detection rates across different studies.

## 4. Discussion

Certain oncogenic viruses play a critical role in the pathogenesis of human malignancies, as they trigger cellular transformation and tumorigenesis through distinct molecular mechanisms [3]. Among these oncoviruses, human papillomavirus (HPV) is a widely prevalent DNA virus that is capable of infecting the oral mucosae of both HIV-positive individuals and the general population, significantly contributing to HPV-associated carcinogenesis. In recent years, a notable increase in the number of oropharyngeal squamous cell carcinoma (OPSCC) cases has been recorded [35,36]. HIV-related immunosuppression compromises the host’s capacity to eliminate HPV infections, resulting in their persistence. Numerous studies have described a significant rise in both the prevalence and diversity of human papillomavirus (HPV) genotypes detected in the oral mucosae of HIV-positive individuals compared to the general population [37,38,39]. Pugliese et al. reported that HR-HPV-31 was the most frequently detected genotype, followed by HPV-16 and HPV-18, and they also observed that there had been a gradual increase in the prevalence of LR-HPV over time [40]. Moreover, they reported that there was no statistically significant association between the prevalence of oral HPV and several demographic or behavioral risk factors, including age, gender, HIV status, substance use, or sexual history. Conversely, Biala et al. described a strong association between high-risk oral HPV genotypes and HIV-related immunodeficiency, with HPV-16 and HPV-18 being more prevalent in the oral mucosae of individuals with lower CD4 counts. Meanwhile, HPV-18, HPV-33, and HPV-52 showed significant correlations with higher HIV RNA levels and severe immunosuppression, emphasizing the role of immune status in oral HPV’s persistence and progression [11]. Bottolico et al. demonstrated that the oral cavity served as a reservoir for a diverse spectrum of HPV genotypes, with Alphapapillomavirus (α-HPV) emerging as the most prevalent genus (62.4%) among HIV-positive individuals, followed by Betapapillomavirus (β-HPV) (28.2%) and Gammapapillomavirus (γ-HPV) (9.4%) [12]. These findings underscore the coexistence of both mucosal and cutaneous HPV types within the oral microenvironment, highlighting the broad and complex landscape of HPV colonization and the potential transmission pathways. Fatahzadeh et al. identified oncogenic α-HPV genotypes (HPV-16, HPV-18, and HPV-31) in 23% of their participants, emphasizing their potential role in oropharyngeal carcinogenesis [41]. Additionally, the detection of non-oncogenic α-HPV types (HPV-6, HPV-11) in 40% of cases suggests an association with benign oral lesions, reinforcing the notion of the diverse clinical implications of oral HPV infections. Garcia et al., identified HPV-18 as the most prevalent high-risk genotype (24.1%), highlighting its significant implications in the pathogenesis of head and neck cancer [31]. Morais et al., found that HPV DNA was present in 65.9% of cervical, 63.8% of anal, and 4.2% of oral samples, with higher HIV viral loads (≥75 copies/mL) significantly associated with HPV presence [42]. Wood et al., reported a low prevalence (3.6%) of oral/oropharyngeal HPV, consistent with global and South African data, and found no significant association between the presence of HPV and several behavioral and demographic factors [43]. Nemcova et al. detected HPV infections in 96.8% of anal and 23.6% of oral samples, with HPV-16 being the most common HR-HPV genotype in both anal (25.4%) and oral (2.5%) cases [44]. These studies collectively highlight the high prevalence and variability of HPV infections among HIV-positive individuals, emphasizing the influence of immunosuppression, sexual behavior, and the anatomical site on the persistence of HPV. HPV-16 and HPV-18 consistently emerge as the most frequently detected high-risk genotypes, particularly in the anal and oropharyngeal regions, reinforcing their strong association with HPV-related malignancies. In this study, the high prevalence and heterogeneity of HPV-58 (6.23%), HPV-16 (4.326%), and HPV-66 (3.733%) indicated that there was significant variability across populations and methodologies, supporting their association with HPV-related oropharyngeal malignancies and the need for the continuous surveillance of HIV-positive individuals. Similarly, the elevated detection of LR-HPV genotypes, particularly HPV-13 (7.16%), HPV-5 (5.64%), and HPV-62 (4.24%), suggests that there is substantial heterogeneity, which is likely influenced by patients’ immune statuses, geographic variations, and differences in detection methods. Based on these results, the potential presence of publication bias should be acknowledged, as studies with negative or non-significant findings are less likely to be published. Moreover, language restrictions may have led to the exclusion of relevant research, potentially affecting the comprehensiveness of the review. Additionally, the lack of distinction between high-risk (HR) and low-risk (LR) HPV genotypes in many studies may limit the evaluation of their differential clinical significance and oncogenic potential, thereby impacting the robustness of the findings.

Given the high heterogeneity observed in the prevalence of HPV genotypes among HIV-positive individuals, future studies may benefit from the use of advanced analytical methods such as federated learning (FL) [45]. Since FL enables secure, multi-center data analysis while preserving patients’ privacy, it is a promising approach to the integration of diverse datasets without direct data sharing. Additionally, FL’s privacy-preserving and scalable framework could enhance global HPV surveillance efforts by facilitating cross-institutional collaboration and ensuring compliance with data protection regulations. Furthermore, its integration with IoT devices and predictive analytics may contribute to personalized risk assessments and the early detection of HPV-related complications, particularly in immunocompromised populations. Researchers have also emphasized the value of large-scale pharmacovigilance analyses in HPV prevalence research, particularly in immunocompromised populations [46]. Integrating real-world data could improve the assessment of HPV vaccines and their antiviral safety. As seen in anti-VEGF studies, future research should explore HPV’s long-term risks in HIV-positive individuals so as to optimize prevention strategies and contribute to safer HPV interventions.

This study has several limitations that should be considered when interpreting the findings. Firstly, the heterogeneity in the data sources may affect the comparability of the results, as the studies included in this analysis differed in terms of their populations’ characteristics, their geographic distributions, and the diagnostic methods used for HPV detection. Secondly, the lack of distinction between high-risk (HR) and low-risk (LR) HPV genotypes in many excluded studies prevented a comprehensive evaluation of their differential clinical significance and oncogenic potential. Despite these limitations, this study provides valuable insights into the prevalence of HPV genotypes among HIV-positive individuals. Future research should aim to address these limitations by incorporating standardized diagnostic approaches, larger and more diverse cohorts, and longitudinal data, which would enhance the accuracy and clinical applicability of HPV epidemiological studies.

## 5. Conclusions

The results of this study demonstrate the influence of HIV-associated immunosuppression on the persistence of HPV infections, emphasizing its significant role as a risk factor for HPV-related oropharyngeal and head and neck cancers. Given the critical involvement of HPV in oral carcinogenesis, further research is warranted to develop targeted preventive and therapeutic strategies tailored to this vulnerable population. This systematic meta-analysis, which focuses on high-risk HPV genotypes in the oral mucosae, offers critical insights into the persistence of HPV in HIV-positive individuals. It highlights HPV’s role in oncogenesis and the development of effective intervention strategies. The evaluation of the oral mucosae in HIV-positive individuals is of critical importance for the early detection of oral lesions, dysplasia, and oropharyngeal cancers associated with the persistence and progression of HPV infections. Our findings suggest that there is significant heterogeneity in the prevalence of LR-HPV genotypes in HIV-positive individuals, with some genotypes showing higher detection rates across studies. The presence of broad confidence intervals for several genotypes further emphasizes the variability in the detection methods, study populations, and sample sizes. In conclusion, our results highlight the critical importance of continuous HPV surveillance and targeted vaccination strategies in HIV-positive populations, particularly considering the persistence and recurrence of LR-HPV infections in immunocompromised individuals, while acknowledging potential biases and methodological limitations. The marked heterogeneity in the prevalence of HPV genotypes points to the necessity of employing advanced data analysis methods to enhance HPV monitoring and implement effective control measures in this vulnerable population.

## Figures and Tables

**Figure 1 microorganisms-13-00646-f001:**
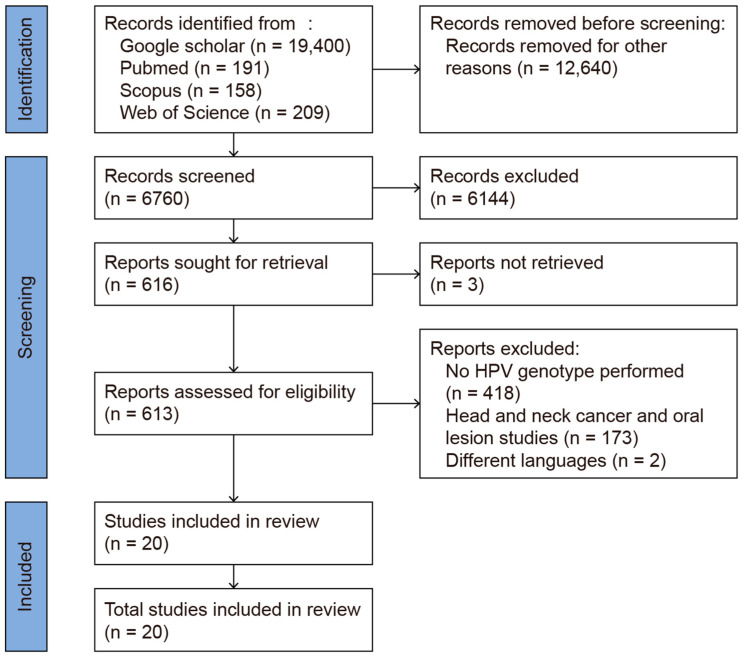
PRISMA 2020 flow diagram of the database search.

**Figure 2 microorganisms-13-00646-f002:**
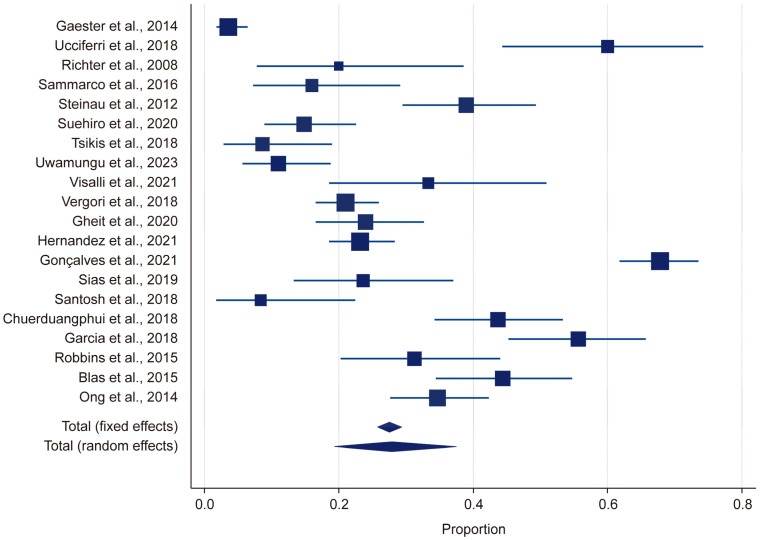
Forest plot: HPV [13,14,15,18,19,20,21,22,23,24,25,26,27,28,29,30,31,32,33,34].

**Figure 3 microorganisms-13-00646-f003:**
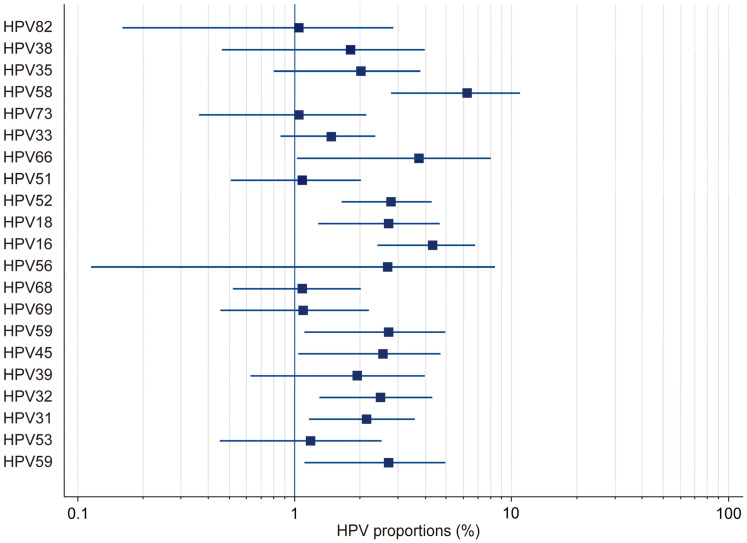
Forest plot: total HR-HPV.

**Figure 4 microorganisms-13-00646-f004:**
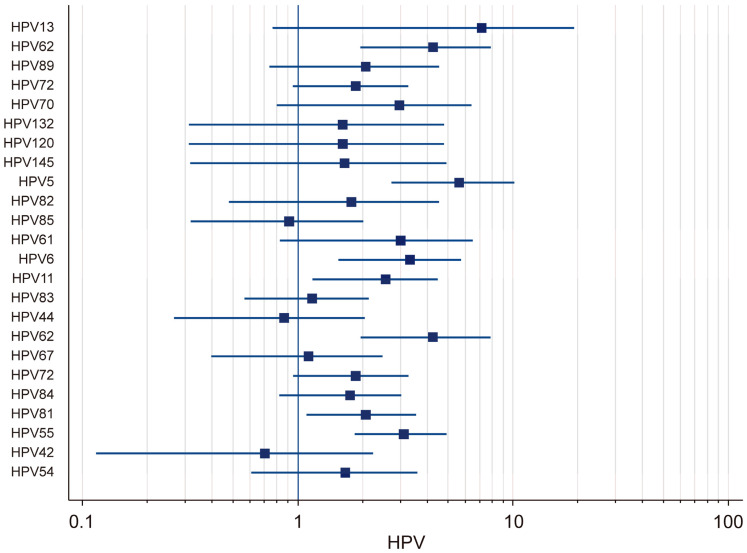
Forest plot: total LR-HPV.

**Table 1 microorganisms-13-00646-t001:** Meta-analysis results: total HPV proportions.

Study	Sample Size	Proportion (%)	95% CI	Weight (%) Random
Gaester et al., 2014 [13]	283	3.53	1.71 to 6.41	5.23
Ucciferri et al., 2018 [14]	45	60	44.33 to 74.31	4.80
Richter et al., 2008 [19]	30	20	7.71 to 38.57	4.59
Sammarco et al., 2016 [15]	50	16	7.14 to 29.11	4.85
Steinau et al., 2012 [20]	100	39	29.40 to 49.27	5.07
Suehiro et al., 2020 [21]	115	14.78	8.85 to 22.61	5.10
Tsikis et al., 2018 [22]	58	8.62	2.86 to 18.98	4.91
Uwamungu et al., 2023 [23]	100	11	5.62 to 18.83	5.07
Visalli et al., 2021 [24]	36	33.33	18.56 to 50.97	4.69
Vergori et al., 2018 [25]	305	20.98	16.55 to 25.99	5.23
Gheit et al., 2020 [26]	117	23.93	16.53 to 32.69	5.10
Hernandez et al., 2021 [27]	302	23.18	18.54 to 28.35	5.23
Gonçalves et al., 2021 [18]	255	67.84	61.73 to 73.53	5.22
Sias et al., 2019 [28]	55	23.64	13.23 to 37.02	4.89
Santosh et al., 2018 [29]	36	8.33	1.75 to 22.47	4.69
Chuerduangphui et al., 2018 [30]	110	43.64	34.21 to 53.42	5.09
Garcia et al., 2018 [31]	97	55.67	45.23 to 65.76	5.06
Robbins et al., 2015 [32]	64	31.25	20.24 to 44.06	4.94
Blas et al., 2015 [33]	99	44.44	34.45 to 54.78	5.07
Ong et al., 2014 [34]	173	34.68	27.62 to 42.28	5.17
Total (random effects)	2430	27.97	19.34 to 37.51	100

**Table 2 microorganisms-13-00646-t002:** Meta-analysis results: HR-HPV proportions.

Study	Sample Size	Proportion (%)	95% CI	Weight (%) Random
Gaester et al., 2014 [13]	283	3.53	1.71 to 6.41	5.19
Ucciferri et al., 2018 [14]	45	57.78	42.15 to 74.34	4.84
Richter et al., 2008 [19]	30	20	7.71 to 38.57	4.65
Sammarco et al., 2016 [15]	50	16	7.14 to 29.11	4.87
Steinau et al., 2012 [20]	100	39	29.40 to 49.27	5.06
Suehiro et al., 2020 [21]	115	14.78	8.85 to 22.61	5.08
Tsikis et al., 2018 [22]	58	8.62	2.86 to 18.98	4.92
Uwamungu et al., 2023 [23]	100	11	5.62 to 18.83	5.06
Visalli et al., 2021 [24]	36	33.33	18.56 to 50.97	4.74
Vergori et al., 2018 [25]	305	20.98	16.55 to 25.99	5.19
Gheit et al., 2020 [26]	117	23.93	16.53 to 32.69	5.09
Hernandez et al., 2021 [27]	302	5.63	3.13 to 8.86	5.19
Gonçalves et al., 2021 [18]	255	67.84	61.73 to 73.53	5.18
Sias et al., 2019 [28]	55	23.64	13.23 to 37.02	4.91
Santosh et al., 2018 [29]	36	8.33	1.75 to 22.47	4.74
Chuerduangphui et al., 2018 [30]	110	43.64	34.21 to 53.42	5.08
Garcia et al., 2018 [31]	97	55.67	45.23 to 65.76	5.05
Robbins et al., 2015 [32]	64	31.25	20.24 to 44.06	4.95
Blas et al., 2015 [33]	99	44.44	34.45 to 54.78	5.06
Ong et al., 2014 [34]	173	34.68	27.62 to 42.28	5.14
Total (random effects)	2430	26.65	17.43 to 37.04	100

**Table 3 microorganisms-13-00646-t003:** Meta-analysis results: LR-HPV proportions.

Study	Sample Size	Proportion (%)	95% CI	Weight (%) Random
Gaester et al., 2014 [13]	283	3534	1707 to 6402	5.19
Ucciferri et al., 2018 [14]	45	57,778	42,150 to 72,343	4.84
Richter et al., 2008 [19]	30	20	7714 to 38,567	4.65
Sammarco et al., 2016 [15]	50	16	7170 to 29,113	4.87
Steinau et al., 2012 [20]	100	39	29,401 to 49,269	5.06
Suehiro et al., 2020 [21]	115	14,783	8854 to 22,610	5.08
Tsikis et al., 2018 [22]	58	8621	2859 to 18,983	4.92
Uwamungu et al., 2023 [23]	100	11	5621 to 18,830	5.06
Visalli et al., 2021 [24]	36	33,333	18,556 to 50,970	4.74
Vergori et al., 2018 [25]	305	20,984	16,551 to 25,989	5.19
Gheit et al., 2020 [26]	117	23,932	16,529 to 32,698	5.09
Hernandez et al., 2021 [27]	302	5629	3313 to 8860	5.19
Gonçalves et al., 2021 [18]	255	67,843	61,731 to 73,534	5.18
Sias et al., 2019 [28]	55	23,636	13,228 to 37,020	4.91
Santosh et al., 2018 [29]	36	8333	1753 to 22,469	4.74
Chuerduangphui et al., 2018 [30]	110	43,636	34,202 to 53,422	5.08
Garcia et al., 2018 [31]	97	55,67	45,227 to 65,757	5.05
Robbins et al., 2015 [32]	64	31,25	20,242 to 44,059	4.95
Blas et al., 2015 [33]	99	44,444	34,454 to 54,776	5.06
Ong et al., 2014 [34]	173	34,682	27,620 to 42,280	5.14
Total (random effects)	2430	26,654	17,428 to 37,043	100

## Data Availability

No new data were created or analyzed in this study.

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
