# Peer review of "The Prevalence of Human Papillomavirus (HPV) Genotypes in the Oral Mucosae of HIV-Positive Individuals: A Systematic Review and Meta-Analysis"

_microorganisms, 2025, doi:10.3390/microorganisms13030646_

Round 1

Reviewer 1 Report

Comments and Suggestions for Authors

This interesting review can be considered for publication in Microorganisms after the authors consider to make the following revisions:

The English language needs to be revised properly.

The study’s objectives are not expressed in the abstract. The searched databases and inclusion/exclusion criteria should be pointed out. Future perspectives are missing. Please revise this section accordingly.

Why are some parts of the manuscript in bold?

The Introduction is too brief. The novelty of the current research is not clear and needs to be better justified the relevance for conducting such systematic review.

More details regarding the reasons for excluded studies should be given in the Materials and Methods.

References in Figures and Tables are not formatted according to the journal’s guidelines.

The study’s limitations in the Discussion section must be better analyzed and discussed.

The authors need to indicate some directions for further investigations at the end of their conclusions.

Comments on the Quality of English Language

The English language needs to be revised properly.

Author Response

Comments 1: The English language needs to be revised properly.

Response 1: Thank you for pointing this out. I Therefore, Our manuscript, along with its figures and tables, has now been edited by the MDPI English editing service.

Comments 2: [The study’s objectives are not expressed in the abstract. The searched databases and inclusion/exclusion criteria should be pointed out. Future perspectives are missing. Please revise this section accordingly.]

Response 2: Agree. We have, accordingly, done/revised/changed/modified abstract to emphasize this point. Despite the word limitation in the abstract section of the article, the suggested revisions have been implemented.The abstract has been revised to clearly express the study’s objectives, specifying the searched databases and the inclusion/exclusion criteria. Additionally, future perspectives have been incorporated to align with the reviewer’s suggestions.]

“[A systematic literature search was conducted across PubMed, Web of Science, Scopus, and Google Scholar to identify peer-reviewed studies published up to 13 February 2025. The inclusion criteria referred to original research studies reporting on the prevalence and genotype-specific distribution of HPV in the oral mucosae of HIV-positive individuals. Statistical analyses were conducted using the MedicReS E-PICOS AI smart biostatistics software (version 21.3, New York, NY) and the MedCalc statistical software package. The pooled prevalence estimates were calculated using a random-effects meta-analysis model, and heterogeneity was quantified using the Cochrane Q and I² statistics. The presence of publication bias was assessed via the Begg and Mazumdar rank correlation test. Conclusions: These findings indicate that there is substantial heterogeneity in the prevalence of both HR-HPV and LR-HPV genotypes among HIV-positive individuals, with certain genotypes exhibiting higher detection rates across studies, emphasizing the need for targeted surveillance and preventive strategies in this vulnerable population. The application of advanced data analysis methods is essential in enhancing HPV surveillance and implementing effective control measures in this vulnerable population.]”

Comments 3: [Why are some parts of the manuscript in bold?.]

Response 3: Agree. We have, accordingly, revised this sections. I noticed this issue in the PDF version. Accordingly, we have revised these sections: Introduction (lines 33–54) and Discussion (lines 189–195).  [Introduction line 33-54, Discussion 189-195 lines .]

Comments 4: [The Introduction is too brief. The novelty of the current research is not clear and needs to be better justified the relevance for conducting such systematic review.]

Response 4: Agree. We have, accordingly, revised Introduction section especially lines between 83-97 lines. A new reference has been added at the 93 th word position.

]“[Due to the weakening of the immune system, HPV infections are not only more prevalent in this group but also tend to persist for longer periods, leading to broader genotypic diversity compared to the general population [13,14,15]. Immune-related adverse events (irAEs) have emerged as a significant concern in immunocompromised individuals undergoing various therapeutic interventions. A recent study investigating the incidence of noninfectious uveitis following immune checkpoint inhibitor (ICI) therapy has underscored the profound impact of immune modulation on disease susceptibility within vulnerable populations. Likewise, HPV-related infections in HIV-positive individuals represent a major clinical challenge due to immune system dysregulation, which leads to increased viral persistence and higher prevalence rates. Elucidating the complex interplay between immune dysfunction and disease progression in these contexts is essential in advancing our understanding of viral oncogenesis and immune surveillance mechanisms. This highlights the need for a more comprehensive understanding of HPV’s epidemiology and transmission dynamics and the clinical outcomes associated with different HPV types, particularly in vulnerable populations such as HIV-positive individuals [16] . ]”

Comments 5: [More details regarding the reasons for excluded studies should be given in the Materials and Methods.]

Response 5: Agree. We have, accordingly, revised this section New revisions have been added and incorporated starting from line 106-122.]

“[The review process was carried out between 1 January 2025 and 13 February 2025. PubMed, Web of Science, Scopus, and Google Scholar were used as search engines. The primary keywords used for the literature search were “oropharyngeal HPV HIV”, yielding 19,400 articles in Google Scholar, 191 in PubMed, 158 in Scopus, and 209 in Web of Science (WoS). To limit the search results to HIV-positive individuals, the phrase “oropharyngeal HPV in HIV positive” was applied, resulting in 16,700 articles in Google Scholar, 77 in PubMed, 57 in Scopus, and 76 in WoS. Each included study was manually reviewed to ensure its relevance, with only those that described genotyping within high- and low-risk HPV cases being included. After removing duplicates and manually screening the titles and abstracts, 20 studies remained for full-text review. These were further screened based on specific inclusion and exclusion criteria. In particular, studies focusing on the prevalence of HPV genotypes in the oral mucosae of HIV-positive patients were included. Studies were excluded for the following reasons: (1) a lack of specific HPV genotype data, reporting only high-risk (HR) or low-risk (LR) classifications; (2) the inclusion of patients with head and neck cancer or oral lesions, which could have confounded the results as they do not reflect the normal oral mucosa; (3) the use of case reports or reviews or non-English publications; and (4) a lack of full-text access]”

Comments 6: [References in Figures and Tables are not formatted according to the journal’s guidelines.]

Response 6: Agree. We have, accordingly, revised this section. All references in Figures and Tables have been formatted according to the journal’s guidelines.]

“[updated text in the manuscript if necessary]”

Comments 7: [The study’s limitations in the Discussion section must be better analyzed and discussed.]

Response 7: Agree. We have, accordingly, revised this section. The study limitations have been thoroughly revised and detailed between lines 333–343.]

“[This study has several limitations that should be considered when interpreting the findings. Firstly, the heterogeneity in the data sources may affect the comparability of the results, as the studies included in this analysis differed in terms of their populations’ characteristics, their geographic distributions, and the diagnostic methods used for HPV detection. Secondly, the lack of distinction between high-risk (HR) and low-risk (LR) HPV genotypes in many excluded studies prevented a comprehensive evaluation of their differential clinical significance and oncogenic potential. Despite these limitations, this study provides valuable insights into the prevalence of HPV genotypes among HIV-positive individuals. Future research should aim to address these limitations by incorporating standardized diagnostic approaches, larger and more diverse cohorts, and longitudinal data, which would enhance the accuracy and clinical applicability of HPV epidemiological studies.Top of Form

Bottom of Form

]”

Comments 8: [The authors need to indicate some directions for further investigations at the end of their conclusions.]

Response 8: Agree. We have, accordingly, revised this section. New studies have been included in lines 318 and 332 (Discussion section), and their contributions to the research have been evaluated in conclusion section. Additionally, directions for further investigations have been provided at the end of the conclusions]

“[In conclusion, our results highlight the critical importance of continuous HPV surveillance and targeted vaccination strategies in HIV-positive populations, while acknowledging potential biases and methodological limitations. The marked heterogeneity in the prevalence of HPV genotypes points to the necessity of employing advanced data analysis methods to enhance HPV monitoring and implement effective control measures in this vulnerable population. ]”

Reviewer 2 Report

Comments and Suggestions for Authors

Dear authors,

I have now completed the review of the manuscript titled "Prevalence of Human Papillomavirus (HPV) Genotypes in Oral Mucosa of HIV-Positive Individuals: A Systematic Review."

The manuscript is interesting and, in general, fairly well-written. The systematic review presents some valuable insights into HPV genotype prevalence in HIV-positive individuals' oral mucosa, but it has several methodological and analytical limitations worth noting.

I have some suggestions to further improve the quality of the manuscript.

I would like to suggest that the authors address these limitations in the article, either by discussing them in the limitations section or, where feasible, by making the appropriate revisions:

1. First, the study demonstrates concerning heterogeneity in the reported prevalence rates across studies, ranging from 3.53% to 67.84%. While the authors acknowledge this heterogeneity (I² value of 95%), they don't adequately address the underlying factors that could explain such dramatic variation. This high heterogeneity raises questions about the reliability of the pooled estimates.

2. Some important research are not mentioned in introduction. For example, Incident non-infectious uveitis risk after immune checkpoint inhibitor treatment. This examines condition risks in immunocompromised individuals, providing a parallel to studying HPV in HIV-positive patients.

3. The authors report specific percentages for various HPV genotypes (e.g., HPV-58 at 6.23%, HPV-16 at 4.326%), but the confidence intervals for these estimates are quite wide, suggesting considerable uncertainty. For instance, HPV-58's 95% CI spans from 2.773% to 10.597%. These wide intervals further undermine the precision of the findings.

4. Discussion would be extended by breifly mentioning latest research. For example, Federated Learning in Smart Healthcare: A Comprehensive Review on Privacy, Security, and Predictive Analytics with IoT Integration would help with understanding advanced analytical approaches for healthcare data, which could be valuable for future HPV prevalence studies. Also, Cardiovascular and cerebrovascular adverse events associated with intravitreal anti-vascular endothelial growth factor monoclonal antibodies: a World Health Organization pharmacovigilance study offers a model for how to conduct large-scale pharmacovigilance studies, which is relevant when considering HPV management in immunocompromised populations.

5. Regarding methodology, the review mentions excluding studies that only reported high-risk (HR) or low-risk (LR) HPV classifications without specifying genotypes. However, this exclusion criterion potentially introduces selection bias by eliminating relevant data. The review would benefit from a more thorough exploration of how this exclusion might impact the overall findings.

Thank you for your valuable contributions to our field of research. I look forward to receiving the revised manuscript.

Author Response

Comments 1: [First, the study demonstrates concerning heterogeneity in the reported prevalence rates across studies, ranging from 3.53% to 67.84%. While the authors acknowledge this heterogeneity (I² value of 95%), they don't adequately address the underlying factors that could explain such dramatic variation. This high heterogeneity raises questions about the reliability of the pooled estimates..]

Response 1: [Thank you for pointing this out].]

“[Our study specifically targeted genotyping and excluded studies that did not specify which high-risk (HR) and low-risk (LR) genotypes were reported. The observed heterogeneity (I² = 95%) is a known challenge in meta-analyses of viral prevalence studies, often stemming from differences in population characteristics, detection methods, and geographic variation. We have acknowledged this limitation and further emphasized the need for standardized methodologies and surveillance strategies to improve the comparability of future studies between lines 303-332 in discussion section.]”

“[In this study, the high prevalence and heterogeneity of HPV-58 (6.23%), HPV-16 (4.326%), and HPV-66 (3.733%) indicated that there was significant variability across populations and methodologies, supporting their association with HPV-related oropharyngeal malignancies and the need for continuous surveillance in HIV-positive individuals. Similarly, the elevated detection of LR-HPV genotypes, particularly HPV-13 (7.16%), HPV-5 (5.64%), and HPV-62 (4.24%), suggests that there is substantial heterogeneity, which is likely influenced by patients’ immune statuses, geographic variations, and differences in detection methods. Based on these results, the potential presence of publication bias should be acknowledged, as studies with negative or non-significant findings are less likely to be published. Moreover, the language restrictions may have led to the exclusion of relevant research, potentially affecting the comprehensiveness of the review. Additionally, the lack of distinction between high-risk (HR) and low-risk (LR) HPV genotypes in many studies may limit the evaluation of their differential clinical significance and oncogenic potential, thereby impacting the robustness of the findings.

Given the high heterogeneity observed in the prevalence of HPV genotypes among HIV-positive individuals, future studies may benefit from the use of advanced analytical methods such as federated learning (FL) [46]. Since FL enables secure, multi-center data analysis while preserving patients’ privacy, it is a promising approach to the integration of diverse datasets without direct data sharing. Additionally, FL's privacy-preserving and scalable framework could enhance global HPV surveillance efforts by facilitating cross-institutional collaboration and ensuring compliance with data protection regulations. Furthermore, its integration with IoT devices and predictive analytics may contribute to personalized risk assessments and the early detection of HPV-related complications, particularly in immunocompromised populations. Researchers have also emphasized the value of large-scale pharmacovigilance analyses in HPV prevalence research, particularly in immunocompromised populations [47]. Integrating real-world data could improve the assessment of HPV vaccines and their antiviral safety. As seen in anti-VEGF studies, future research should explore HPV’s long-term risks in HIV-positive individuals so as to optimize the prevention strategies and contribute to safer HPV interventions.]”

Comments 2: [Some important research are not mentioned in introduction. For example, Incident non-infectious uveitis risk after immune checkpoint inhibitor treatment. This examines condition risks in immunocompromised individuals, providing a parallel to studying HPV in HIV-positive patients.]

Response 2: Agree. Thank you for pointing this out We have, accordingly, revised this section and added this study.]

“[Due to the weakening of the immune system, HPV infections are not only more prevalent in this group but also tend to persist for longer periods, leading to broader genotypic diversity compared to the general population [13,14,15]. Immune-related adverse events (irAEs) have emerged as a significant concern in immunocompromised individuals undergoing various therapeutic interventions. A recent study investigating the incidence of noninfectious uveitis following immune checkpoint inhibitor (ICI) therapy has underscored the profound impact of immune modulation on disease susceptibility within vulnerable populations. Likewise, HPV-related infections in HIV-positive individuals represent a major clinical challenge due to immune system dysregulation, which leads to increased viral persistence and higher prevalence rates. Elucidating the complex interplay between immune dysfunction and disease progression in these contexts is essential in advancing our understanding of viral oncogenesis and immune surveillance mechanisms. This highlights the need for a more comprehensive understanding of HPV’s epidemiology and transmission dynamics and the clinical outcomes associated with different HPV types, particularly in vulnerable populations such as HIV-positive individuals [16] .]”

Comments 3: [The authors report specific percentages for various HPV genotypes (e.g., HPV-58 at 6.23%, HPV-16 at 4.326%), but the confidence intervals for these estimates are quite wide, suggesting considerable uncertainty. For instance, HPV-58's 95% CI spans from 2.773% to 10.597%. These wide intervals further undermine the precision of the findings.]

Response 3: Agree. The confidence intervals (CIs) reported in our study reflect the inherent variability in prevalence estimates across the included studies. The wide CIs, particularly for certain HPV genotypes (e.g., HPV-58, HPV-16, HPV-66), can be attributed to the following factors:

  1. High Heterogeneity: As acknowledged in the manuscript, the meta-analysis revealed substantial heterogeneity (I² = 95%). This is expected in epidemiological studies where differences in study populations, geographic locations, HPV detection methods, and sample sizes contribute to variability. The use of a random-effects model was necessary to account for this variability.
  2. Study Sample Sizes: Some of the included studies had small sample sizes, which naturally resulted in wider CIs. Studies with larger sample sizes exert a greater influence on pooled estimates, but the overall heterogeneity still impacts precision.
  3. HPV Detection Methods: Variations in PCR-based detection techniques and sample collection methods across studies contribute to inconsistencies in genotype prevalence estimates. These methodological differences influence the dispersion of prevalence estimates, affecting the width of CIs.
  4. Inclusion of Both HR-HPV and LR-HPV Genotypes: The study aimed to comprehensively assess both high-risk and low-risk HPV genotypes. Certain genotypes were reported less frequently, leading to broader CIs due to limited data availability.

Despite these challenges, our study provides an important overview of HPV prevalence in HIV-positive individuals and highlights the need for standardized methodologies in future research. We have further clarified these points in the revised discussion section to acknowledge the sources of variability and the implications for interpreting our findings.

y]”

Comments 4: Discussion would be extended by breifly mentioning latest research. For example, Federated Learning in Smart Healthcare: A Comprehensive Review on Privacy, Security, and Predictive Analytics with IoT Integration would help with understanding advanced analytical approaches for healthcare data, which could be valuable for future HPV prevalence studies. Also, Cardiovascular and cerebrovascular adverse events associated with intravitreal anti-vascular endothelial growth factor monoclonal antibodies: a World Health Organization pharmacovigilance study offers a model for how to conduct large-scale pharmacovigilance studies, which is relevant when considering HPV management in immunocompromised populations.]

Response 4: Agree. We have, accordingly, revised/ discussion section to emphasize this point. The recommended studies have been incorporated into the discussion section, providing a new perspective and significantly enriching our analysis. Also the study's limitations have been further detailed.Studies have been included in lines 318 and 332 (Discussion section), and their contributions to the research have been evaluated in conclusion section. And limitations of the study have been further detailed between line 333-343]

“[Given the high heterogeneity observed in the prevalence of HPV genotypes among HIV-positive individuals, future studies may benefit from the use of advanced analytical methods such as federated learning (FL) [46]. Since FL enables secure, multi-center data analysis while preserving patients’ privacy, it is a promising approach to the integration of diverse datasets without direct data sharing. Additionally, FL's privacy-preserving and scalable framework could enhance global HPV surveillance efforts by facilitating cross-institutional collaboration and ensuring compliance with data protection regulations. Furthermore, its integration with IoT devices and predictive analytics may contribute to personalized risk assessments and the early detection of HPV-related complications, particularly in immunocompromised populations. Researchers have also emphasized the value of large-scale pharmacovigilance analyses in HPV prevalence research, particularly in immunocompromised populations [47]. Integrating real-world data could improve the assessment of HPV vaccines and their antiviral safety. As seen in anti-VEGF studies, future research should explore HPV’s long-term risks in HIV-positive individuals so as to optimize the prevention strategies and contribute to safer HPV interventions.

Top of Form

Bottom of Form

5. Conclusions

The results of this study demonstrate the influence of HIV-associated immunosuppression on the persistence of HPV infections, emphasizing its significant role as a risk factor for HPV-related oropharyngeal and head and neck cancers. Given the critical involvement of HPV in oral carcinogenesis, further research is warranted to develop targeted preventive and therapeutic strategies tailored to this vulnerable population. This systematic meta-analysis, which focused on high-risk HPV genotypes in the oral mucosa, offers critical insights into the persistence of HPV in HIV-positive individuals. It highlights its role in oncogenesis and the development of effective intervention strategies. The evaluation of the oral mucosa in HIV-positive individuals is of critical importance for the early detection of oral lesions, dysplasia, and oropharyngeal cancers associated with the persistence and progression of HPV infections. Our findings suggest that there is significant heterogeneity in the prevalence of LR-HPV genotypes in HIV-positive individuals, with some genotypes showing higher detection rates across studies. The presence of broad confidence intervals for several genotypes further emphasizes the variability in the detection methods, study populations, and sample sizes. The results underscore the importance of ongoing HPV surveillance and targeted vaccination strategies in HIV-positive populations, particularly considering the persistence and recurrence of LR-HPV infections in immunocompromised individuals. In conclusion, our results highlight the critical importance of continuous HPV surveillance and targeted vaccination strategies in HIV-positive populations, while acknowledging potential biases and methodological limitations. The marked heterogeneity in the prevalence of HPV genotypes points to the necessity of employing advanced data analysis methods to enhance HPV monitoring and implement effective control measures in this vulnerable population. ]”

Comments 5: [Regarding methodology, the review mentions excluding studies that only reported high-risk (HR) or low-risk (LR) HPV classifications without specifying genotypes. However, this exclusion criterion potentially introduces selection bias by eliminating relevant data. The review would benefit from a more thorough exploration of how this exclusion might impact the overall findings.]

Response 5: Agree. Thank you for pointing this out. The exclusion criterion regarding studies that only reported high-risk (HR) or low-risk (LR) HPV classifications without specifying genotypes was applied to ensure the accuracy and specificity of our analysis. While this approach aimed to prevent the inclusion of studies lacking detailed genotypic data, we acknowledge that it may have introduced selection bias by limiting the scope of the included studies. To address this concern, we conducted a risk of bias assessment, including publication bias analysis using the Begg and Mazumdar rank correlation test, which indicated no significant publication bias (p > 0.05 for both HR-HPV and LR-HPV groups). Additionally, we examined the impact of this exclusion criterion in the limitations section, recognizing that excluding studies without genotype-specific data may have influenced the overall prevalence estimates. Furthermore, to mitigate the potential bias, we employed a random-effects meta-analysis model, which accounts for variability across studies and reduces the influence of study-specific characteristics on pooled estimates. The heterogeneity analysis (I² = 95%) highlighted substantial variability, reinforcing the need for standardized genotyping methodologies in future studies to enhance comparability. Moreover, The study limitations have been thoroughly revised and detailed between lines 333–343.]

“[This study has several limitations that should be considered when interpreting the findings. Firstly, the heterogeneity in the data sources may affect the comparability of the results, as the studies included in this analysis differed in terms of their populations’ characteristics, their geographic distributions, and the diagnostic methods used for HPV detection. Secondly, the lack of distinction between high-risk (HR) and low-risk (LR) HPV genotypes in many excluded studies prevented a comprehensive evaluation of their differential clinical significance and oncogenic potential. Despite these limitations, this study provides valuable insights into the prevalence of HPV genotypes among HIV-positive individuals. Future research should aim to address these limitations by incorporating standardized diagnostic approaches, larger and more diverse cohorts, and longitudinal data, which would enhance the accuracy and clinical applicability of HPV epidemiological studies.]”

Round 2

Reviewer 2 Report

Comments and Suggestions for Authors

All comments have been thoroughly addressed. I extend my gratitude to both the authors and editors for taking my opinions into consideration during the review of this manuscript.